# Training Interpretable Convolutional Neural Networks towards Class-Specific Filters

## Abstract

Convolutional neural networks (CNNs) have often been treated as "black-box" and successfully used in a range of tasks. However, CNNs still suffer from the problem of filter ambiguity – an intricate *many-to-many correspondence* relationship between filters and features, which undermines the models' interpretability. To interpret CNNs, most existing works attempt to interpret a pre-trained model, while neglecting to reduce the filter ambiguity hidden behind. To this end, we propose a simple but effective strategy for training interpretable CNNs. Specifically, we propose a novel Label Sensitive Gate (LSG) structure to enable the model to learn disentangled filters in a supervised manner, in which redundant channels experience a periodical shutdown as flowing through a learnable gate varying with input labels. To reduce redundant filters during training, LSG is constrained with a sparsity regularization. In this way, such training strategy imposes each filter's attention to just one or few classes, namely class-specific. Extensive experiments demonstrate the fabulous performance of our method in generating sparse and highly class-related representation of the input. Moreover, comparing to the standard training strategy, our model displays less redundancy and stronger interpretability.

## 1 Introduction

Convolutional Neural Networks (CNNs) demonstrate extraordinary performance in various visual tasks (Krizhevsky et al., 2012; He et al., 2016; Girshick, 2015; He et al., 2017a). However, the strong expressive power of CNNs is still far from interpretable, which significantly limits its applications that require humans' trust or interaction, e.g. self-driving and medical image analysis (Caruana et al., 2015; Bojarski et al., 2017). In this paper, we argue that *filter ambiguity* is one of the most critical reasons that hampers the interpretability of CNNs. As a matter of fact, previous studies has shown that 1) filters in CNNs generally extract features of a mixture of various semantic concepts, including objects, parts, scenes, textures, materials and colors (Zhang et al., 2018b; Bau et al., 2017); and that 2) there is also redundant overlap between features extracted by different filters (Prakash et al., 2019). The intricate *many-to-many correspondence* relationship between filters and features is so-called filter ambiguity as shown on the left of Figure 1.

Obviously, in high convolutional layers which might capture class-related feature, filter ambiguity contradicts our intention of an interpretable CNN, because it hinders humans from interpreting the concepts of a filter (Zhang et al., 2018b), which has been shown as an essential role in the visualization and analysis of networks (Olah et al., 2018) in human-machine collaborative systems (Zhang et al., 2017a;c). Moreover, the unnecessary overlap between features extracted by different filters leads to under-utilization of a model's expressiveness (Prakash et al., 2019) Therefore, reducing filter ambiguity is critical to obtain better feature with better interpretability and less redundancy.

However, it is non-trivial to achieve such a goal barricaded by substantial challenges. First, most interpretability-related research simply focuses on post-hoc interpretation of filters (Szegedy et al., 2013; Bau et al., 2017), which manages to interpret the main semantic concepts captured by a filter but fails to alleviate the filter ambiguity prevalent in pretrained models. Second, many existing works such as VAEs' variants (Higgins et al., 2017; Burgess et al., 2018; Kim & Mnih, 2018; Chen et al., 2018; Kumar et al., 2017) and InfoGAN (Chen et al., 2016) try to disentangle data representation and obtain better interpretability in an unsupervised way. However, it is proved that unsupervised

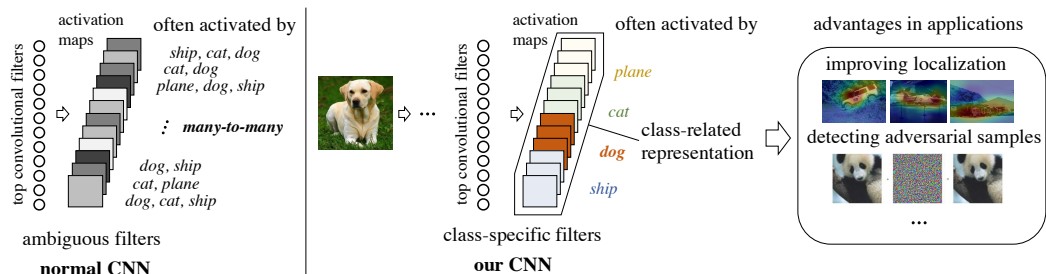

Figure 1: The motivation of learning class-specific filters. In a normal CNN, each filter corresponds to multiple classes, since it extracts a mixture of features from different classes Zhang et al. (2018b), which is a symptom of filter ambiguity. In contrast, we enforce each filter to correspond to one (or few) classes, namely to be class-specific, which brings better interpretability and class-related feature representation. Such representation not only facilitates understanding the inner logic of CNNs, but also improves different application tasks related to the mechanism of filters, which is verified by our exhaustive experiments.

learning on disentangled features without inductive bias is impossible (Locatello et al., 2018), which challenges the works above.

Considering the aforementioned challenges, we shed light on enforcing the one-class for one-filter relationship during training (instead of post-hoc) in a supervised manner only with classification labels. To this end, we propose a novel training strategy that coerces each filter into extracting features from only one or few classes for classification tasks, namely *disentangling filters towards class-specific*. Specifically, we design a Label Sensitive Gate (LSG) structure on the top of convolutional filters, which limits each filter's activation only to its specific input label(s). In our training process, we periodically insert LSG into the CNN and jointly minimize the classification cross-entropy and the sparsity of LSG, so as to keep the model's performance on classification and meanwhile encourage class-specific filters.

Experiments proved the LGS filter yield better representation for images hence leading to promising applications. Our training method makes data representation sparse and highly correlated with the labeled class, which not only illustrates the alleviation of filter ambiguity but also enhances the interpretability of the network. The advantages of our method are concretely substantiated by sparser correlation between filters, sparser filter-class correlation, better explanation for misclassification and more precise localization of labeled objects.

**Contributions** The contributions of this work can be summarized as: (1) we propose a novel training strategy for CNNs which forces each filter to extract features mainly from only one or a few classes; (2) moreover, we propose a metric to evaluate filter ambiguity, which can also be used as a regularization in our training to encourage class-specific filters; (3) finally, our training enables filters to output sparse and class-related representation, which helps alleviate filter redundancy, improve interpretability of the model.

## 2 RELATED WORKS

Existing works related to our work include post-hoc filter interpretation, learning disentangled representation and model pruning.

**Post-hoc Interpretation for Filters** is widely studied, which aims to interpret the patterns captured by filters in pretrained CNNs. Plenty of works visualize the pattern of a neuron as an image, which is the gradient (Zeiler & Fergus, 2014; Mahendran & Vedaldi, 2015; Simonyan et al., 2013) or accumulated gradient (Mordvintsev et al., 2015; Olah et al., 2018) of a certain score about the activation of the neuron. Bau et al. (2017) determine the main visual patterns extracted by a convolutional filter by treating it as a pattern detector. Some other works transfer the representation in CNN into an explanatory graph (Zhang et al., 2017b; 2018a) or a decision tree (Zhang et al., 2019), which aim to figure out the visual patterns of filters and the relationship between co-activated patterns. Post-hoc filter interpretation helps to understand the main patterns of a filter but makes no change to the existing filter ambiguity of the pretrained models, while our work aims to train interpretable models.

**Learning Disentangled Representation** refers to learning data representation that encodes different semantic information into different dimensions. As a principle, it proves impossible to learn disentangled representation without inductive bias (Locatello et al., 2018). Unsupervised methods such as variants of VAEs (Kingma & Welling, 2014) and InfoGAN (Chen et al., 2016) rely on regularization. VAEs (Kingma & Welling, 2014) are modified into many variants (Higgins et al., 2017; Burgess et al., 2018; Kim & Mnih, 2018; Chen et al., 2018; Kumar et al., 2017), while their disentangling performance is sensitive to hyperparameters and random seeds. InfoGAN (Chen et al., 2016) learns disentangled hidden representations by maximizing the mutual information between input and generated images. Some other unsupervised methods rely on special network architectures including interpretable CNNs (Zhang et al., 2018b) and CapsNet (Sabour et al., 2017). As for supervised methods, Thomas et al. (2018) propose to disentangle with interaction with the environment; Bouchacourt et al. (2018) apply weak supervision from grouping information, while our work applies weak supervision from classification labels.

**Model Pruning** reduces structure redundancy and therefore is widely used for both model efficiency and better generalization, while our work uses it to alleviate filter ambiguity and improve interpretability. Many existing methods prune pretrained CNNs for better efficiency without significant deterioration in performance, including channel-wise pruning (He et al., 2017b; Molchanov et al., 2016; Liu et al., 2017) and filter-wise pruning (Luo et al., 2017). The methods mentioned imply that CNNs trained by standard training strategy tend to have redundant filters. Further research exploits redundant filters to improve generalization and suppress over-fitting by temporally pruning during training: Dense-Sparse-Dense (Han et al., 2016) encourages sparse weights by regularization and then recovers dense weights by removing the regularization; RePr (Prakash et al., 2019) repeatedly prunes and reinitializes redundant filters to reduce filter overlap.

## 3 METHOD

Learning disentangled filters in CNNs alleviates filter ambiguity and meanwhile narrows the gap between human's perception and CNN's representations. In this section, we first present an ideal case of class-specific filters, which is a direction for our training, and then we elaborate on our method about how to train an interpretable network.

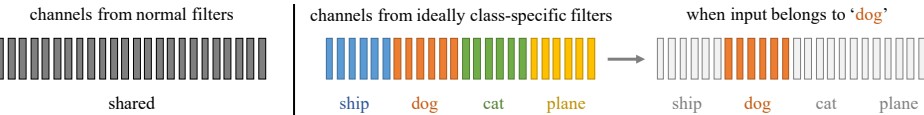

Figure 2: The intuition of disentangling filters to class-specific. In a normal CNN, each filter extracts a mixture of features from different classes (Zhang et al., 2018b), which is a symptom of filter ambiguity. In contrast, when filters are ideally class-specific, each filter extracts features mainly from only one class. Given the labeled class of an input, the filters irrelevant to the class have weak activation, and hence classification performance changes little when they are shut down.

### 3.1 CLASS-SPECIFIC FILTERS

For CNNs in classification tasks, it is demonstrated that the filters in the last convolutional layer extract high-level features related to certain classes more or less (Zeiler & Fergus, 2014). We suppose the most ideal case, as shown in Figure 2, is that each filter extracts features mainly from only one class, i.e. each filter is mainly relevant to one class. We call such filters ideally class-specific. Obviously, given the labeled class of an input, the classification performance changes little if we shut down the channels from the filters irrelevant to the class. This is the direction for our training method elaborated later.

To have a rigorous definition of "ideally class-specific", for a CNN, we use a matrix $G \in \{0,1\}^{C \times K}$ to measure the relevance between filters and classes, where $K$ is the number of filters, $C$ is the number of classes, and each element $G_c^k$ represents the relevance between the $k$-th filter and $c$-th class, as shown in Figure 3. The $k$-th filter extracts features mainly in the $c$-th class when and only when $G_c^k = 1$. Given the label $y \in \{1, 2, ..., C\}$ of an input, we can index a row $G_y \in \{0,1\}^K$ from the matrix $G$, which can be used as a gate multiplied to the feature maps to shut down those irrelevant

channels. We call filters in a CNN as *ideally class-specific* filters, when the network's prediction $\tilde{y}^G$ approaches that of original network $\tilde{y}$ after the feature maps from the last convolutional layer are multiplied by the gate $G_y$.

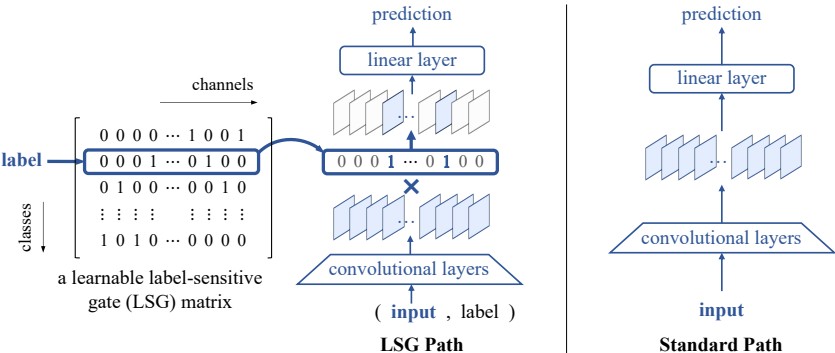

Figure 3: The framework of Label-Sensitive Gate (LSG) training. During LSG training, we *alternately* train a CNN through both the LSG path and the standard (STD) path. In the STD path, network parameters are optimized to minimize the cross-entropy. In the LSG path, feature maps after the last convolutional layer pass through the learnable gate which is a row vector in the LSG matrix indexed by the label of the input. Network parameters and LSG matrix are optimized to minimize the cross-entropy in conjunction with a sparsity regularization for the LSG matrix. When testing, we just run the STD path.

## 3.2 PROBLEM FORMULATION

In order to train a CNN towards disentangling filters to class-specific and meanwhile keep the original classification performance, as opposed to the standard training scheme that entangles filters, we introduce a Label-Sensitive Gate (LSG) path in addition to the standard path of forward propagation. In such a LSG path, some channels are shut down by multiplication with a learnable gate. This path's classification performance is regarded as a regularization for disentanglement training.

As shown in Figure 3, let us denote $\theta$ as the network parameters. The network forward propagates in two paths: 1) the standard (STD) path predicting $\tilde{y}_\theta$, and 2) the LSG path with gate matrix $G$ predicting $\tilde{y}_\theta^G$ where the learnable gate $G_y$ for the input labeled with $y$ is multiplied to the feature map before the linear layer.

Initially, the elements in the gate matrix $G$ for a CNN are unknown. Intuitively, we can search for them by exploring the binary space and find one solution that yields the best classification performance through LSG path, i.e., to solve $\Phi_0(\theta) = \min_{G:G^k \text{is one-hot}} \text{CE}(y||\tilde{y}_\theta^G)$, where the constraint on $G^k$ applies to $\forall k \in \{1, 2, ..., K\}$ and $\Phi_0$ evaluates the performance of the CNN with filters disentangled. Therefore, it is natural to add $\Phi_0$ into training loss as a regularization that forces filters to be class-specific. Thus, we get the following formulation of the original problem to train a CNN towards ideally class-specific filters as

$$\min_\theta L_0(\theta) = \text{CE}(y||\tilde{y}_\theta) + \lambda\Phi_0(\theta). \tag{1}$$

However, the problem is difficult to solve in practice. On the on hand, the assumption that each filter is ideally disentangled to extract only one class hardly holds, since it is usual for several classes to share one high-level feature in CNNs; on the other hand, binary vectors in a non-continuous space are difficult to optimize with gradient descent.

To address these two issues, we relax the constraints from two perspectives. First, we relax the one-hot vector $G^k$ to sparse vector by $L_1$ regularization $\|G\|_1$ where at least one element equals to 1 ($\|G^k\|_\infty = 1$). Since $\|G^k\|_\infty = 1$ ensures $\|G\|_1$ to be larger than $K$, we actually apply $|\|G\|_1 - s|$ as the sparsity regularization term in order to make the norm of gate matrix close to a constant target norm $s \geq K$. Second, we relax the binary vector $G^k \in \{0, 1\}^C$ to a continuous vector $G^k \in [0, 1]^C$. Therefore $\Phi_0$ now becomes

$$\Phi(\theta) = \min_G\{\text{CE}(y||\tilde{y}_\theta^G) + a|\|G\|_1 - s|\} \quad s.t. \; \|G^k\|_\infty = 1 \text{ and } G_c^k \in [0, 1], \tag{2}$$

where $a$ is a coefficient to balance classification and sparsity. $\Phi$ can be regarded as *a metric of filter ambiguity*, which means a CNN with higher class-specificity of filters will have a lower $\Phi$ (contrary to common sense, here we expect $\Phi$ to be as small as possible).

Replacing $\Phi_0$ in Equation 1 with $\Phi$, we get an intermediate problem $\min_\theta(\mathrm{CE}(y||\tilde{y}_\theta) + \lambda\Phi(\theta))$. It is mathematically equivalent if we move $\min_G$ within $\Phi$ to the left most (see Appendix A for proof). Thus, we formulate a relaxed problem as

$$\min_{\theta,G} L(\theta,G) = \mathrm{CE}(y||\tilde{y}_\theta) + \lambda(\mathrm{CE}(y||\tilde{y}_\theta^G) + a|\,\|G\|_1 - s|), \qquad (3)$$

$$s.t.\ \|G^k\|_\infty = 1 \text{ and } G_c^k \in [0,1].$$

The relaxed problem is easier to solve by jointly optimizing $\theta$ and $G$ with gradient descent, compared to either the discrete optimization in the original problem, or the nested optimization in the intermediate problem. Solving the relaxed problem, we can obtain a CNN for classification with class-specific filters, where $G$ precisely describes the correlation between filters and classes.

### 3.3 OPTIMIZATION

To solve the optimization problem formulated in Equation 3 with gradient descent, we *alternately* optimize $G$ and $\theta$ to improve the classification performance of the model while ensuring the constraints of sparsity on $G$, as is shown in Algorithm 1 [1]. In this scheme, we run the LSG path to update both $G$ and $\theta$ in some epochs of a period, and run the STD path to update $\theta$ in the other epochs. When $G$ is updated with gradient, $G^k$ will be normalized by $\|G^k\|_\infty$ to ensure $\|G^k\|_\infty = 1$, and then clipped into range $[0,1]$.

After solving the relaxed problem, we can further tighten the constraint on LSG to $G \in \{0,1\}^{C \times K}$ for stronger sparsity of $G$. In this work, we tighten into a simple and special case, where some filters are related to only one class (i.e. $G^k$ is one-hot) for class-specific features, and the other filters are related to all classes (i.e. $G^k$ is all-one) for class-sharing features. Specifically, if $G^k$ has only one element $G_c^k \geq h$ (threshold $h \in (0,1]$) $G_k$ will be set to one-hot, otherwise $G_k$ will be set to all-one.

---

**Algorithm 1** LSG Training
___
1: **for** E in epochs **do**
2:  **for** N in batches **do**
3:    **if** E%3 == 1 **then**
4:      $\tilde{y}_\theta^G \leftarrow$ prediction through the LSG path with $G$
5:      $Cost \leftarrow \lambda(\mathrm{CE}(y||\tilde{y}_\theta^G) + a|\,\|G\|_1 - s|)$
6:      $G$ is updated using the gradient decent as $G \leftarrow G - \epsilon\frac{\partial Cost}{\partial G}$;
7:      Each column of $G^k$ is normalized as $G^k \leftarrow \frac{G^k}{\|G^k\|_\infty}$;
8:      $G \leftarrow \mathrm{clip}(G, 0, 1)$
9:    **else**
10:      $\tilde{y}_\theta \leftarrow$ prediction through the STD path
11:      $Cost \leftarrow \mathrm{CE}(y||\tilde{y}_\theta)$
12:    **end if**
13:    $\theta \leftarrow \theta - \epsilon\frac{\partial Cost}{\partial \theta}$
14:  **end for**
15: **end for**

---

## 4 EXPERIMENT

In this section, we conduct five experiments. We first delve into LSG training from three aspects, so as to respectively study the effectiveness of LSG training, the class-specificity of filters and the correlation between filters train with LSG. After that, we demonstrate LSG's application on improving

---

[1]Another choice is a naive scheme: in all epochs we predict through both paths to directly calculate $L(\theta,G)$ defined in equation 3 and update $\theta$ and $G$ with gradients of it. We choose the alternating scheme because in our preliminary exploration it shows better training stability and converging speed.

object localization and adversarial samples detection. In the following parts, we denote our training method Label-Sensitive Gate as *LSG*, the standard training as *STD*, and CNNs trained with them as *LSG CNNs* and *STD CNNs*, respectively.

**Training setting**    We trained ResNet20s (He et al., 2016) on CIFAR20 and ResNet152s (He et al., 2016) on PASCAL VOC 2010 (Everingham et al.) on classification task with LSG/STD training. ResNet20s are evaluated in this 4.1, 4.2, 4.3 and 4.5, and ResNet152s are evaluated in 4.1 and 4.4. For fair comparison, all the code is implemented on the Pytorch framework and tested on GTX 1080Ti GPUs. We report the validation performance on last epoch following common practice.

For ResNet20s, the default settings include: batch size is 256; the optimizer is SGD with momentum of 0.9 (Sutskever et al., 2013); the initial learning rate is 0.1; and the total training epochs is 150. The ResNet152s are finetuned from model pretrained on ImageNet (Deng et al., 2009). Parameters are frozen except the top 2 bottleneck blocks, gate matrix and linear layers. The setting is: batch size is 32; the optimizer is Adam (Kingma & Ba, 2014); the initial learning rate is 1e-5 for STD path and 1e-3 for LSG path; the total training epochs is 150.

Beside, we preprocess PASCAL VOC to be a classification dataset for training ResNet152s: we crop out images for the objects in 6 classes (bird, cat, dog, cow, horse and sheep) and resize the image to 128x128; then randomly reassign 3644 objects for training and 1700 objects for testing. No segmentation label is used in training ResNet152s.

## 4.1    EFFECTIVENESS OF LSG TRAINING

To begin with, we conduct experiments to demonstrate the effectiveness of our LSG training in learning a sparse gate matrix and yielding a low $\Phi$ that evaluates class-specificity of filters.

**Basic Quantitative Evaluation**    To compare the performance on classification and filter disentanglement of the STD CNN and the LSG CNN, we calculate their test accuracy, cross entropy, L1 Norm $\|G\|_1$ and $\Phi$ as mentioned in Section 3.2. As shown in Table 1, LSG CNN's even slightly outperforms STD CNN in test accuracy, and cross entropy of both is comparable, while LSG CNN has much lower L1 Norm and $\Phi$. The L1 Norm and the $\Phi$ indicate LSG learns class-specific filters that hence reduce filter ambiguity. These metrics quantitatively demonstrate LSG's capability of learning a sparse gate matrix and disentangling filters without any sacrifice on classification accuracy.

Table 1: Metrics of the STD CNN and the LSG CNN.

| Dataset | Model | Training | Accuracy | Cross Entropy | L1 Norm | $\Phi$ |
|---------|-------|----------|----------|---------------|---------|--------|
| CIFAR-10 | ResNet20 | LSG (Ours) | **0.9062** | 6.9762 | **0.1742** | **0.2203** |
| | | STD | 0.9046 | **6.8657** | 0.4757 | 0.8234 |
| PASCAL VOC 2010 | ResNet152 | LSG (Ours) | **0.8506** | 0.0508 | **0.1996** | **0.0044** |
| | | STD | 0.8429 | **0.0356** | 0.8488 | 0.2794 |

**Visualization of Label-Sensitive Gate Matrix**    To illustrate the relevance between classes and the learned filters described in the gate matrix, we visualize gate matrices in Figue 4. Subfigure (a) indicates that the LSG training yields a sparse LSG matrix where each filter is only related to one or few classes. Subfigure (b) comes from a LSG training without the sparsity regularization to the gate matrix (L1 norm), from which we observe filter ambiguity where a filter would extract ambiguous features from multiple classes. Accordingly, LSG training effectively learn sparse gate matrix, and this characteristic originates from our sparsity regularization – the L1 norm on the gate matrix.

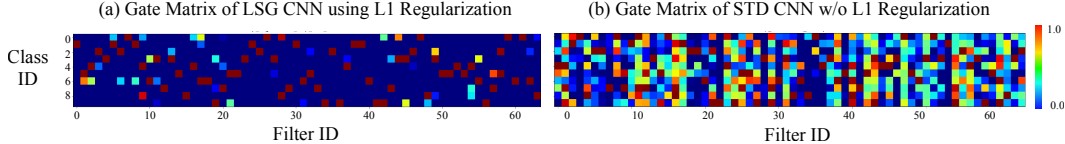

Figure 4: Visualization of the gate matrices from LSG training w/o the sparsity regularization on the gate matrix. The x-axis is the filter id from 1 to 64, the y-axis is the class id in CIFAR10 from 0 to 9, and the color represents how much a filter is related to a class.

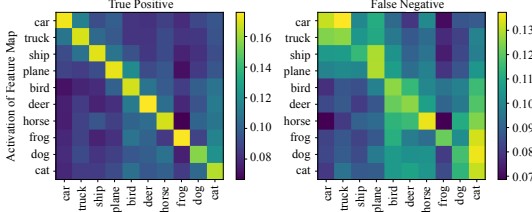

Figure 5: Classification confusion matrix for STD / LSG models when masking filters highly related to the first class (a1, b1), and the first and sixth classes (a2, b2).

## 4.2 CLASS-SPECIFIC FILTERS

To substantiate that LSG training learns class-specific filters and reveal how it works, we explore the LSG CNN with filters masked for certain class(es) from which we observe the similarity between feature vectors and gates. The experiments are conducted on the ResNet20 trained in 4.1.

**Masking Filters for Certain Class**  Although the sparse gate matrix implies filters are class-specific, to further verify the LSG training learns class-specific filters, we remove the filters highly related to certain class(es) referencing the gate matrix (i.e. $G_c^k$ greater than a threshold), and then visualize the classification confusion matrices. As shown in Figure 5, when filters highly related to "plane" are removed, to our surprise, the LSG CNN fails to recognize the first class "plane"; nevertheless, the STD CNN still manages to recognize "plane". Analogously, when masking the filters highly related to the first and sixth classes, we observe similar phenomenon. This demonstrates that in the LSG CNN, the features from a certain class are almost extracted by the specific filters highly related to the class, while other filters extract little feature from the class. Therefore the filters are trained to be class-specific under the encouragement of the sparse gate matrix.

**Similarity Between Features and Gates**  This experiment explains how the LSG matrix encourages class-specific filters by analyzing the similarity between feature from each class and each row in the LSG matrix. In our models there is a global average pooling (GAP) layer after the last convolutional layer, which yields a feature vector denoted as $a$. We use inner production to measure the similarity $S_y^c$ between the feature vectors $a_y$ of images in the class $y$ and the $c$-th row $G_c$ in the LSG matrix (see Appendix B for details). We calculate $S_y^c$ over all true positive (TP) and false negative (FN) images respectively and obtain similarity matrices $S_{TP}, S_{FN} \in \mathbb{R}^{C \times C}$, as shown in Figure 6.

From the figure, we observe two phenomena and provide the following analysis. 1) TP similarity matrix is diagonally dominant significantly, which reveals how LSG works in disentangling filters to class-specific: LSG forces filters to yield feature vectors whose direction approaches that of the gate vector for its related class. 2) FN similarity matrix is far from diagonally dominant. Besides, two classes with shared features, such as car & truck and ship & plane, have high similarity in the FN similarity matrix, which enlightens us that hard samples with ambiguous feature across classes tends to be misclassified in LSG models. Thus, the mechanism of misclassification in the LSG CNN is probably the features across classes extracted by the shared filters .

Figure 6:  The similarity between feature vectors from a class (y-axis) and a row in the LSG matrix (x-axis), averaged over all TP/FN samples. We reorder classes in CIFAR10 for better visualization.

## 4.3 CORRELATION BETWEEN FILTERS

To study the filter correlation in LSG CNNs, we train a CNN with the LSG under a tighter constraint, so as to conveniently analyze the correlation between filters for different classes.

**Constraint Tightening**     Before tightening the constraint on gate matrix $G \in \{0, 1\}^{C \times K}$ according to Section 3.3, we first conduct a statistic analysis on 20 different LSG models to figure out how many filters each class monopolizes. The result indicates that in CIFAR10, each class tends to monopolize 10% total filters (6 for ResNet20's last convolutional layer), and 10% from all filters are shared by classes. Inspired by this phenomenon, we tighten the constraint accordingly by manually setting a fixed LSG matrix. In the fixed LSG matrix, for each row $G_c$ there are 6 ones from 6 columns of one-hot $G^k$, and 4 extra columns $G^k$ are all-one (see Appendix C for illustration). This provides a setup where each class monopolizes 6 filters and 4 extra filters are shared by all classes. Models trained with this constraint naturally inherits all features of previous LSG models, and moreover, has better disentangled filters.

**Filter Orthogonality Analysis**     We evaluate the correlation between filters with the orthogonality of their weights. We train an AlexNet (Krizhevsky et al., 2012) and a ResNet20 with the fixed LSG matrix or with STD, and then calculate the correlation between filters in each models. The correlation between two filters is defined as the inner product of their normalized weights, visualized as matrices in Figure 7. In subfigures (a), (c) for STD models, the filters are randomly correlated with each other. On the contrary, subfigures (b), (d) for LSG models, the matrices are approximately block-diagonal, which means the correlation between the filters is limited into several filter groups corresponding to classes. This implies that filters for the same class are highly correlated (non-orthogonal) for the co-occurrence of features extracted by them, while filters for different classes are almost uncorrelated (orthogonal) for the lack of co-occurrence.

Through intuitive reasoning, we can explain why LSG training encourages filters for different classes to become orthogonal. Given a class $c$ and a gate matrix, that assigns the filter $k$ for class $c$ and filter $k'$ for other class. During training, filter $k'$ is closed (i.e. its activation is masked) in the LSG path when class $c$'s images input. In order to ensure the STD path predict similar to LSG path, the filter $k'$ tends to be activated by class $c$ as less as possible, which implies the weight of filter $k'$ is approximately perpendicular to $V_c$, the linear space spanned by class $c$'s features in a layer before. The filter $k$ for class $c$, however, tends to be activated by class $c$ as saliently as possible so as to enable the CNN to recognize this class. So the weight of filter $k$ is approximately within $V_c$. Overall, the weights of filter $k$ and filter $k'$ tends to be orthogonal.

**Filter Redundancy**     With further analysis based on filter correlation, we discover that LSG reduce the redundancy of filters. For each element $a_{i,j}$ in the correlation matrix, we define that the i-th filter and j-th filter are correlated if $a_{i,j} > s$ where $s$ is a varying threshold. We count the ratio of elements in the correlation matrix satisfying the constrain and plot the results in subfigure (e). It shows that LSG significantly reduces the redundant weak (e.g. threshold$= 0.3$) correlation between most filter pairs, which are mostly for different classes. We explain the reduction of filter redundancy as a natural consequence of encouraging filters (for different classes) to be orthogonal. For a set of filter groups orthogonal to each other, a filter in any group can not be a linear representation with the filers from other groups. This directly avoids redundant filter across groups. Besides, experiments in (Prakash et al., 2019) also verify the opinion that filter orthogonality reduces filter redundancy.

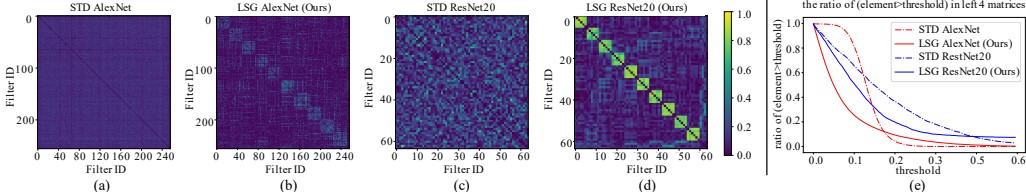

Figure 7: The correlation (inner product) matrix of filters in AlexNet and ResNet20 trained with STD/LSG. (e) shows the ratio of each matrix' elements larger than a varying threshold.

**Class-Related Representation**     Based on filter correlation, we can even show that filters trained with LSG can yield highly class-related representation, namely the representation for an image tends to exactly correspond to its labeled class rather than to other classes. That's based on a reasoning that in the highly activated channel of a representation, filters are less activated by other classes and less correlated to the filters for other classes. To verify this reasoning, we analyze the correlation between the filters highly activated by each class. First we pick out each class $c$'s dominant $m$ filters, denoted as group $A_c$ (class $c$ has highest average activation on them; $m = 25$ for AlexNet; $m = 6$ for RestNet20). We define Inter-Class filter correlation as the average correlation between

filters in different classes' group: $C_{\text{IC}} = \frac{1}{C(C-1)m^2} \sum_c \sum_{c \neq c'} \sum_{k \in A_c} \sum_{k' \in A_{c'}} C_{k,k'}$, where $C_{k,k'}$ is the row $k$ column $k'$ of the filter correlation matrix in Figure 7. The results in Table 2 show that inter-class filter correlation in LSG is about half as much as that in STD, both on AlexNet and ResNet20. This demonstrates that different classes tend to activate uncorrelated filters in LSG CNNs. As a result, the representations for different classes have less overlap. Thus we finally confirm representation from LSG CNN is highly class-related.

Table 2: Inter-Class Filter Correlation ($C_{\text{IC}}$)

| Model | STD AlexNet | LSG AlexNet (Ours) | STD ResNet20 | LSG ResNet20 (Ours) |
|---|---|---|---|---|
| $C_{\text{IC}}$ | 0.1224 | **0.0684** | 0.1648 | **0.1022** |

## 4.4 APPLICATION - LOCALIZATION

In this subsection, we exploit our class-specific filters to improve the localization of CNNs. LSG training encourages top convolutional each filter to focus on fewer classes, which obviously implies its feature map can localize a class better. We conducts experiments below to verify it.

**Localization method**     Resizing feature maps to input size is a widely used method to determine the area of objects or visual concepts, which not only works in localization task without bounding box labels (Bau et al., 2017), but also take an important role in network visualization and understanding the function of filters (Zhou et al., 2016).

We study LSG CNNs' performs on localizing object classes with the feature map from both single filter and all filters. For single filter, we bilinear interpolate its feature map and segment the region with values above the top 20% activation of the filter on the entire test dataset. For all filers, we sums up their feature maps with the weights of the linear connections between each channels and an output class[2]. By bilinear interpolating the sum feature map to input size, we get a classification activation map (CAM) (Bau et al., 2017). We segment the region with values above the top 20% activation in a CAM. Thus we can get segmentation map for a class from feature maps.

**Quantitative evaluation**     We train ResNet152s to do classification on preprocessed PASCAL VOC and use Avg-IoU (average intersection over union) and AP15 (average precision 15%) to evaluate their localization for each class and all classes. Higher metrics shows better localization. Especially, for localization with one filter, we report Avg-IoU and AP15 averaged over all filters. See Appendix E for detailed definition of the metrics. The results for localization with one or all filters are shown in Table 3. For localization with one filter, most classes is localized better with LSG CNN. That's because filters trained with LSG tends be activated by the labeled class rather than many other classes, which alleviates other classes' interference on feature maps. Therefore, as a weighted sum of better one-filter feature maps, LSG CNN's CAMs also outperform LSG CNN's.

Table 3: The performance of localization with resized feature maps in the LSG/STD CNN. For almost all classes, LSG CNN significantly outperforms STD CNN both on Avg-IoU and AP15.

| localized with | metric | training | bird | cat | dog | cow | horse | sheep | total |
|---|---|---|---|---|---|---|---|---|---|
| one filter's feature maps | Avg-IoU | LSG | **0.232** | **0.359** | **0.383** | **0.335** | **0.212** | **0.298** | **0.343** |
| | | STD | 0.150 | 0.145 | 0.138 | 0.147 | 0.152 | 0.151 | 0.146 |
| | AP15 | LSG | **0.689** | **0.939** | **0.961** | **0.923** | **0.650** | **0.852** | **0.918** |
| | | STD | 0.447 | 0.429 | 0.401 | 0.434 | 0.458 | 0.452 | 0.432 |
| all filters' feature maps i.e. CAMs | Avg-IoU | LSG | **0.239** | **0.272** | **0.243** | **0.227** | **0.192** | 0.156 | **0.228** |
| | | STD | 0.174 | 0.110 | 0.115 | 0.161 | 0.175 | **0.194** | 0.147 |
| | AP15 | LSG | **0.730** | **0.873** | **0.798** | **0.690** | **0.650** | 0.512 | **0.732** |
| | | STD | 0.505 | 0.280 | 0.307 | 0.503 | 0.607 | **0.630** | 0.438 |

**Visualiziation**     Besides the quantitative evaluation above, in Figure 8, we also visualize sample some images and their CAMs from the STD/LSG CNN. We observe that the CAMs of STD CNN often activate extra or other semantic areas unrelated to the labeled class. However, LSG training

---

[2]It only works for CNNs with global average pooling and one linear layer the last convolutional layer.

successfully helps the CNN find a more precise area of the labeled objects in an image. Such a phenomenon vividly demonstrates that LSG training improves the performance in localization.

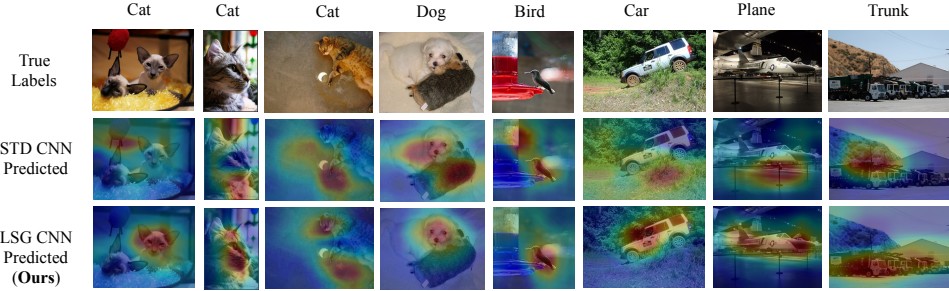

Figure 8: Visualizing the localization in STD CNN and LSG CNN with CAM (Zhou et al., 2016).

## 4.5 APPLICATION - ADVERSARIAL SAMPLE DETECTION

This subsection shows an application of the highly class-related representation from our class-specific filters. It is studied (Wang et al., 2018) that adversarial samples can be detected based on the anomalous behavior of their representation on each layers: in low layers of a neural network, the representation of an adversarial sample is similar to the original class, while on the high layers it is similar to the target class. With the highly class-related representation from our LSG training, it will be much easier to distinguish which class the high-layer representation is similar to. Therefore, we suppose this advantage will help in adversarial detection task.

To verify this judgment, we train a binary classifier with the features of normal samples and adversarial samples extracted by global average pooling after each convolution layers of ResNet20 trained in 4.1. As indicated in (Wang et al., 2018), the non-targeted adversarial samples result in semantic-closer class with the original class, which are hard to discriminate. In this regard, we generate targeted adversarial samples by commonly used PGD (Madry et al., 2017) attack with $\epsilon = 0.3$ and $iter = 40$ and the adversarial target classes are from a random permutation of original classes besides each image's true class. In our experiments, we adopt random forest (Breiman, 2001) as the binary classifier. We randomly select 100, 500 and 1000 images for each class in CIFAR-10 dataset to form different sizes of training datasets while the test data is collected by randomly selecting 100 images for each class. The experimental results are shown in Table 4. We repeat each experiments five times and report the mean AUC scores. The experimental results demonstrate that the class-related representation can better distinguish the abnormal behavior of adversarial samples which can help improve the robustness of the model.

Table 4: The mean Area-Under-Curve (AUC) score for random forest on adversarial detection with features of STD CNN and LSG CNN. With the features of LSG CNN, we can achieve higher AUC score compared with using features of STD CNN, which indicates that the highly class-specific representation makes it easier to distinguish the abnormal behaviour of adversarial samples.

| Num. of training samples | 100 | 500 | 1000 |
|---|---|---|---|
| STD | 72.86 | 80.77 | 83.64 |
| LSG (Ours) | **77.39** | **85.01** | **86.93** |

## 5 CONCLUSION

In this work, we propose a simply yet effective structure – Label Sensitive Gate Matrix to disentangle the filters in CNNs. With reasonable assumptions about the behaviors of filters, we derive regularization terms to constrain the form of the gate matrix. As a result, the sparsity of the gate matrix encourages class-specific filters, and therefore yields sparse and highly class-related representations, which endows model with better interpretability. We believe LSG is a promising architecture to disentangle filters in CNNs. Referring to LSG's successful utility and feasibility in classification problem, we expect that LSG also have the potential to interpret other tasks like detection, segmentation etc, and networks more than CNNs, which is the direction of our future work.

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

# A EQUIVALENCE PROOF

This section is a supplementary proof for the equivalence (mentioned in 3.2) between

$$\min_{\theta}(\text{CE}(y||\tilde{y}_{\theta}) + \lambda\Phi(\theta)),$$

$$\Phi(\theta) = \min_{G}\text{CE}(y||\tilde{y}_{\theta}^{G}) + a\left|\,\|G\|_{1} - s\right| \quad s.t. \ \left\|G^{k}\right\|_{\infty} = 1, G_{c}^{k} \in [0,1],$$

and

$$\min_{\theta,G}L(\theta, G) = \text{CE}(y||\tilde{y}_{\theta}) + \lambda(\text{CE}(y||\tilde{y}_{\theta}^{G}) + a\left|\,\|G\|_{1} - s\right|) \quad s.t. \ \left\|G^{k}\right\|_{\infty} = 1, G_{c}^{k} \in [0,1].$$

This equivalence is true referencing the Lemma below, if we set $x \in X$ in the Lemma as $\theta \in \mathrm{R}^{M}$, $y \in Y$ as $G \in \{G \in [0,1]^{C \times K}\}$, $f(x)$ as $\text{CE}(y||\tilde{y}_{\theta})$, and $g(x, y)$ as $\text{CE}(y||\tilde{y}_{\theta}^{G}) + a\left|\,\|G\|_{1} - s\right|$. Here, $M$ is the number of parameters in a CNN, $C$ is number of classes, $K$ is the number of filters. $y$ is the label of an input, $\tilde{y}_{\theta}$ is the prediction of the CNN's prediction through the STD path, and $\tilde{y}_{\theta}^{G}$ is the prediction of the CNN's prediction through the LSG path with $G$.

**Lemma** Given sets $X, Y$, and functions $f : X \mapsto \mathrm{R}$, $g : X \times Y \mapsto \mathrm{R}$. $f$. If $f$ and $g$ have lower bounds, then

$$\min_{(x,y)\in X\times Y}[f(x) + g(x, y)] = \min_{x\in X}[f(x) + \min_{y\in Y} g(x, y)] \tag{4}$$

*Proof.* Given $\forall x \in X$, $f(x)$ is a constant if we take $y$ as the only variable, so

$$f(x) + \min_{y\in Y} g(x, y) = \min_{y\in Y}[f(x) + g(x, y)]$$

Denote $F(x, y) = f(x) + g(x, y)$, thus to prove equation 4 we only need to prove

$$\min_{(x,y)\in X\times Y} F(x, y) = \min_{x}\min_{y} F(x, y)$$

This is obvious, because for $\forall x \in X, y \in Y$,

$$\min_{(x,y)\in X\times Y} F(x, y) \leq F(x, y),$$

so for $\forall x \in X$

$$\min_{(x,y)\in X\times Y} F(x, y) \leq \min_{y\in Y} F(x, y),$$

and then

$$\min_{(x,y)\in X\times Y} F(x, y) \leq \min_{x\in X}\min_{y\in X} F(x, y).$$

Reversely, for $\forall x \in X, y \in Y$, we have

$$\min_{x\in X}\min_{y\in Y} F(x, y) \leq F(x, y),$$

so

$$\min_{x\in X}\min_{y\in Y} F(x, y) \leq \min_{(x,y)\in X\times Y} F(x, y). \qquad \square$$

## B SIMILARITY BETWEEN FEATURE VECTORS AND GATES

We design a similarity between the feature vector $a_y$ of a image in class $y$ and the $c$-th row $G_c$ in the LSG matrix. Noticing $a, G_c \in \mathbb{R}^K$, we use their inner production $a_y^T G_c$ as a similarity in direction. Thus, each pair of image in class $y$ and class $c$ will contribute an inner production $a_y^T G_c$ to $S_y^c$ – the average similarity between the feature vectors for class $y$ and the $c$-th gate row. Thus, we can further define the average similarity between the feature vectors for class $y$ and the $c$-th gate row,

$$S_y^c(D) = \mathcal{D}_{\text{test}} mean a_y^T G_c : (x, y) \in D.$$

Here, $D$ is a dataset to calculate the average on, and $(x, y)$ is a pair of image and label in the dataset. We can take $D$ as all true positive or all false negative data, as conducted in 4.2.

## C MANUALLY FIXED GATE MATRIX

In 4.3 we use a manually fixed gate matrix to train CNNs from scratch. The gate matrix is visualized in Figure 9.

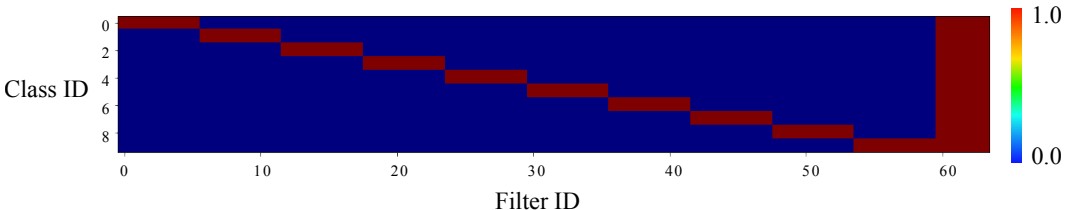

Figure 9: Manual fixed gate matrix. Each class monopolizes 6 filters and 4 extra filters are shared by all classes.

## D CLUSTER CENTER EXPERIMENTS

Using the model with fixed LSG mentioned in 4.3, we run k-means clustering on the feature vectors after the global average pooling in the STD CNN and the LSG CNN. The dataset we used is CIFAR-10. We find the LSG CNN yields better clustering centers, which is almost the same as the gate matrix we used (visualized in Figure 9), with filter groups reordered.

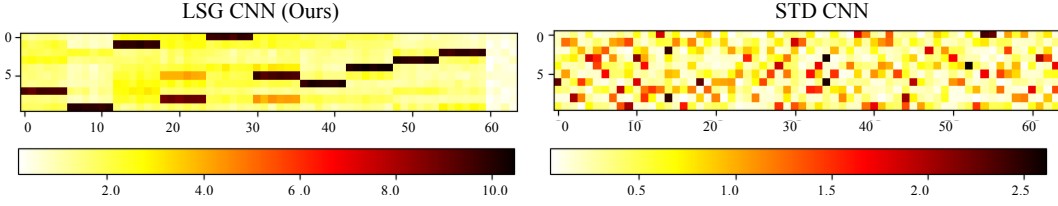

Figure 10: Cluster center experiment. The x-axis is the channel id, and the y-axis is class id. Each row is a the mean of a cluster in the feature vectors' space. and the color represents the value of an element in the mean.

## E METRICS FOR LOCALIZATION

This section gives detailed definition of the metrics we used in our localization experiments.

### E.1 LOCALIZATION WITH ONE FILTER

For an image $x$ in class $c$ (denoted as $x \subset D_c \in D$, $D$ is the dataset), we denote the ground-truth segmentation map for $x$ as $S_x$, and the segmentation map given by the resized feature map from filter $k$ as $\hat{S}_x^k$. The filter $k \in [1, K]$, where $K$ is the number of filers in the top convolutional layer.

The IoU (intersection over union) on this image is defined as

$$\text{IoU}_x^k = \frac{\left\| S_k \cap \hat{S}_k \right\|}{\left\| S_k \cup \hat{S}_k \right\|}.$$

The Avg-IoU (average intersection over union) for filter $k$ to localize class $c$ is defined as

$$\text{IoU}_c^k = \text{mean}_{x \in D_c} \text{IoU}_x^k.$$

The AP15 (average precision 15%) for filter $k$ to localize class $c$ is defined as

$$\text{AP15}_c^k = \text{mean}_{x \in D_c} \text{I}\{\text{IoU}_x^k) > 0.15\}.$$

When $c = \arg\max_{c'} \text{AP15}_{c'}^k$, we call filter $k$ is for localizing class $c$, denoted as $k \in F_c$, where $F_c$ is the set of filters for localizing class $c$. The performance for filter $k$ is evaluated with $\text{IoU}^k = \text{IoU}_{c^*}^k$ and $\text{AP15}^k = \text{AP15}_{c^*}^k$.

Thus we can define the Avg-IoU and AP15 for localizing class $c$ as

$$\text{IoU}_c = \text{mean}_{k \in F_{c^*}} \text{IoU}^k$$

and

$$\text{AP15}_c = \text{mean}_{k \in F_{c^*}} \text{AP15}^k.$$

Besides, the Avg-IoU and AP15 for localizing all classes is defined as

$$\text{IoU} = \text{mean}_{k \in [1,K]} \text{IoU}^k$$

and

$$\text{AP15} = \text{mean}_{k \in [1,K]} \text{AP15}^k.$$

### E.2 LOCALIZATION WITH ALL FILTERS

For an image $x$ in class $c$ (denoted as $x \in D_c$), we denote the ground-truth segmentation map for $x$ as $S_x$, and the segmentation map given by the resized classification activation map (CAM) (Bau et al., 2017) as $\hat{S}_x$.

The IoU (intersection over union) on this image is defined as

$$\text{IoU}_x = \frac{\left\| S_k \cap \hat{S} \right\|}{\left\| S_k \cup \hat{S} \right\|}.$$

The Avg-IoU (average intersection over union) for localizing class $c$ is defined as

$$\text{IoU}_c = \text{mean}_{x \in D_c} \text{IoU}_x.$$

The AP15 (average precision 15%) for localizing class $c$ is defined as

$$\text{AP15}_c = \text{mean}_{x \in D_c} \text{I}\{\text{IoU}_x) > 0.15\}.$$

Besides, the Avg-IoU and AP15 for localizing all classes is defined as

$$\text{IoU} = \text{mean}_{x \in D} \text{IoU}_x$$

and

$$\text{AP15} = \text{mean}_{x \in D} \text{I}\{\text{IoU}_x) > 0.15\}.$$

# F    DEFENDING ADVERSARIAL SAMPLES

In this experiments, inspired by the highly class-related representation, we further explore LSG CNNs' potential in robustness for adversarial attacks. Two black box attacks are conducted, including one pixel attack (Su et al., 2019) and local search attack (Narodytska & Kasiviswanathan, 2016). They try to fool models according to the model's predicted probability without access to the models' parameters and architectures. From the results shown in Table 5, we find both the attacks gain attack success rates on STD CNN much higher than on LSG CNN. This demonstrates that LSG training also improves robustness to CNNs. We guess the robustness is caused by the increase of within-class distance and the decrease of between-class distance, which requires further verification. Robustness is another valuable characteristic of the highly class-related representation from the class-specific filters.

Table 5: Black Box Attack on STD CNN and LSG CNN

| Attack | Metric | STD CNN | LSG CNN |
|---|---|---|---|
| No Attack | Accuracy | 88.03% | 88.85% |
| Single Pixel Attack | Attack Success Rates | 14.00% | 2.00% |
| Local Search Attack | | 15.00% | 2.00% |

