# OpenReview forum: "Training Interpretable Convolutional Neural Networks towards Class-specific Filters"
_ICLR.cc/2020/Conference — Reject_

### Official Review · AnonReviewer2 · 2019-10-22
**Official Blind Review #2**

**Rating:** 6

**Review:**

Contributions: The paper proposes a novel Label Sensitive Gate (LSG) structure to enable the model to learn disentangled filters in a supervised manner. The novelty of the paper is to introduce the Label-Sensitive Gate path during the training, on top of the standard training path. This encourages the filters to extract class-sensitive features.

The highlight of the experimental results is that by incorporating the LSG structure, the author is able to achieve better localization with CAM because the filters are trained to extract more relevant features. Additionally, filters from different classes are kind of "orthogonal" to each other.

However, I have a few questions for the author:

a) In Figure 5, the author mentions "TP/FN" samples in the caption. However, for the plot at the right, it is titled with "False Positive". Is this a typo or do you talk about something else?

b) In Figure 7, the author shows the correlation matrix of filters. Since the filters belonging to one class are highly correlated (block matrices in plot d), why do not train even less filters for each class? If we train less filters for each class, will that hurt the performance? It might be a good direction on how to interpret filters from one class and how to make intra-class filters more interpretable.

Overall, the novelty of the paper is limited by the scope of the experimental results. Why would people care about interpretability of CNN filters is not explained clearly. The paper shows interpretable filters could help with localization. However, this is the only example in the paper I find useful. It would be better if the author could show any other applications. Moreover, I wish the author could spend more paragraph discussing why filters belonging to different classes tend to be orthogonal to each other. This is not clearly written (Eq(3)?). A more theoretical/intuitive explanation would also make the paper stronger as well.

If the paper could omtroduce more real applications of the proposed LSG model and give more theoretical/intuitive explanations of why the inter-class filters are orthogonal, I would be willing to raise up my scores.


**Experience Assessment:**

I have read many papers in this area.

**Review Assessment: Checking Correctness Of Derivations And Theory:**

I carefully checked the derivations and theory.

**Review Assessment: Checking Correctness Of Experiments:**

I carefully checked the experiments.

**Review Assessment: Thoroughness In Paper Reading:**

I read the paper thoroughly.

---

> ### Author Response · Authors · 2019-11-15
> **Discussing the potential in intra-class filters**
>
> Thanks for your careful reading and valuable suggestion.
>
> Q: Why don’t train fewer filters for each class? Will this hurt the performance?
>
> A: It is reasonable that the highly correlated intra-class filters in Fig 7 indicates that we can train fewer filters for each class. However, the motivation of our paper is to improve the interpretability of CNNs by handling the many-to-many mapping of filters and features as shown in Figure 1. Instead of concerning with the intra-class filters, disentangling the inter-class filters plays an important role in improving such interpretability. Thus, in our paper, we mainly focus on disentangling the filters between classes and did not try to redesign the model architectures.
>
> Q: It might be a good direction on how to interpret filters from one class and how to make the intra-class filters more interpretable.
>
> A: We agree with your opinion that it is a potential direction to reduce the correlation of intra-class filters. On the one hand, it could reduce the redundant filters to get a light-weight network structures like model pruning indicated in the related work. On the other hand,  it could help to train a more efficient model with better generalization as in (Prakash, Aaditya, et al.  2019). Besides, we believe with reduce intra-class filter correlation will bring better interpretability. Based on the current work, we will consider to further interpret intra-class filters by adding explicit regularization or filters pruning.
>
> By the way, thanks for your careful review and discover the typo in Figure 5. We do want to say False Negative. We have corrected it in the revised version.
>
> ---
> Prakash, Aaditya, et al. "RePr: Improved Training of Convolutional Filters." Proceedings of the IEEE Conference on Computer Vision and Pattern Recognition. 2019.

---

> ### Author Response · Authors · 2019-11-15
> **The significance of interpretability and application of class-specific filters**
>
>
> Q: (1) Why would people care about interpretability of CNN filters is not explained clearly. (2) It would be better if the author could show any other applications.
>
> A: Interpretability of deep neural networks (DNNs) is essential for practical applications since it enables users to understand the overall strengths and weaknesses of the models, conveys an understanding of how the models will behave in the future, and how to diagnose and correct potential problems. Please also refer to R1 for review #1 for clear motivation of improving interpretability of CNN filters.
>
> And we have conducted more **quantitative analysis towards localization**, and experiments for **adversarial samples detection** as another application. Please refer to Section 4.4 and 4.5 in our revised version. The advantages of the improved interpretability of our LSG CNNs can be summarized as follows.
>
> 1. LSG training leads to class-specific representation which helps us analyze the inner logic of CNNs and makes it easier to extract class-specific features by checking the learned gate matrix.
>
> 2. As shown in Sec 4.4, the localization performances are consistently boosted in most experiments. LSG training encourages top convolutional each filter to focus on fewer classes, which obviously implies feature map can localize a class better.
>
> 3. For the experiments of adversarial samples detection, we can achieve higher AUC scores with LSG CNN features comparing with STD CNN features. The insight behind is that the activation pattern of adversarial and normal samples is gradually different from lower layer to high layers. Thus the class-specific high-level features make it easier to detect the abnormal behaviour of adversarial samples.
>
> We believe that the better class-specific representation indicates more potential value in many tasks associated with the working mechanism of filters.

---

> ### Author Response · Authors · 2019-11-15
> **Explanation of the orthogonality of inter-class filters**
>
>
> Q: Moreover theoretical/intuitive explanation  about the orthogonality of inter-class filters.
>
> A: We have added more discussion about why the inter-class filters are orthogonal to each other in paragraph of Filter Orthogonality Analysis in Sec 4.3.
>
> Intuitively, when inputting an image of a certain class during training, the filters not belonging to the class are closed by the gate (i.e.  their activation is masked). Our optimization target is to reduce the drop in classification performance brought by the masking operation. This target leads those filters to be activated by the class as weakly as possible. This implies the weight of these filters is approximately perpendicular to the linear subspace $V$ spanned by this class’s features in a layer before. On the contrary, to recognize this class, the filters belongs to the class tend to be activated by the class as greatly as possible. So the weight of this class’s filters is  approximately within the features subspace $V$. Overall, the weights of inter-classes filters intend to be orthogonal. Please refer to the Sec 4.3 of the revised paper for more details.

---

### Official Review · AnonReviewer3 · 2019-10-23
**Official Blind Review #3**

**Rating:** 6

**Review:**

This paper proposed an interesting idea of using Label Sensitive Gate (LSG) structure to enforce models to learn disentangled filters for better interpretability of the DNN model. By periodically training with the sparse LSG structure, the model is forced to extract features from only a few classes. The model is trained efficiently in an alternate fashion (with respect to both the network parameters and the sparse gate matrix.) By disentangling the class-specific filters, the model becomes less redundant and more interpretable.

Overall the paper is well-written and well-organized. I  like the idea of imposing a class-specific gated structure for disentangling the representation. And numerical experiments verify the effectiveness of the proposed method in terms of
1) Improved performance
2) Disentangled representation (a small L1 norm on the gate matrix G)
3) Consistent class activation map for different inputs.

I do have some question though
1) Table 1 also reports the L1 norm and \Phi of a STD CNN. What is the gated matrix G here for a STD CNN?
2) Is it very important to have a constraint $\|G^k\|_{\infty}$ in the model? What if an L-1 norm is used directly?

**Experience Assessment:**

I have read many papers in this area.

**Review Assessment: Checking Correctness Of Derivations And Theory:**

I assessed the sensibility of the derivations and theory.

**Review Assessment: Checking Correctness Of Experiments:**

I assessed the sensibility of the experiments.

**Review Assessment: Thoroughness In Paper Reading:**

I read the paper at least twice and used my best judgement in assessing the paper.

---

> ### Author Response · Authors · 2019-11-15
> **Answering questions about learning the gate**
>
>
> Thanks for your accurate summary of our work.
>
> Q:  What is the gated matrix G for STD CNN in Table 1?
>
> A: By solving optimization problem in Equation (2), we can define and obtain the gate G for an STD CNN pre-trained without gate. The detailed implementation is after finishing training the STD CNN, we insert the gate $G$ after the top convolutional layer of STD CNN. Then we use an LSG training algorithm (Algorithm 1) to search for $G$. Different from training an LSG CNN,  the STD CNN’s the parameters (except $G$) are fixed, and we simply update $G$ in each epoch thus the path with gate (LSG path).
>
> Q: Is it possible to use L-1 norm without $||G^k||_\infty$.
>
> A: The aim of applying L-infinity norm for the model is to enforce the filters to take full charge of at least $k$ ($k>=1$) class. In another word, L-infinity constraint let filters be utilized more efficiently (less useless filters).  Without this L-infinity constraint, according to our early experiments, there will be a slight performance degradation because all elements in G tend to become zero to satisfy the L1 constraint. As a result we got some column in G are exactly zero vector, which caused a waste of filters with those zero columns -- these filter will always be closed by the gate no matter which class of the image is input.

---

### Official Review · AnonReviewer1 · 2019-10-24
**Official Blind Review #1**

**Rating:** 3

**Review:**

This paper proposes a method to improve the interpretability of a convolutional neural network (CNN). The main idea is to force the CNN filters to be class specific, i.e. to be associated to a specific class. This is accomplished by a gating function that enforces filters to be sparsely activated.  This would make the model more interpretable by allowing to check which filters/classes have been activated. Results are evaluated in terms of performance, sparsity of the filters and localization accuracy on CIFAR10.

I lean to reject this paper because I consider the proposed motivations not clear and partially misleading. In the introduction it seems that the idea of enforcing class-specific filters makes sense in general because it avoids filters ambiguity and it reduces redundancy (not clear how). Instead, in the actual implementation, this idea is applied only at the last layer of a CNN. This makes sense, because in a CNN filters need to be shared among classes. It’s an important form of regularization, especially when the amount of training data is reduced.
Additionally, the advantage of enforcing the last layer filters to be class-specific is not clear. To me, instead of evaluating the activation of the classification layer, it is possible to check the activation of the filters of the last convolutional layers. However not much more interpretability is added. Furthermore, the improvement on localization as shown in Fig. 6 is only qualitative as the images could have been cherry-picked and there is no real localization measure.

Additional comments:
In table 1, performance of the proposed training is comparable to a standard training on CIFAR10. However, evaluating the proposed approach on a single dataset and only one network is not enough for a clear evidence. Additionally, the obtained performance is below the standard performance of ResNet on CIFAR10 without a clear reason.
Another way to enforce a similar pattern on the gating function would be by maximizing the mutual information between selected filters and classes.


**Experience Assessment:**

I have published in this field for several years.

**Review Assessment: Checking Correctness Of Derivations And Theory:**

I assessed the sensibility of the derivations and theory.

**Review Assessment: Checking Correctness Of Experiments:**

I assessed the sensibility of the experiments.

**Review Assessment: Thoroughness In Paper Reading:**

I read the paper thoroughly.

---

> ### Author Response · Authors · 2019-11-15
> **The motivation of learning class-specific filters and its advantages**
>
> Thanks for your careful review.
>
> Q: The proposed motivations not clear.
>
> A: Sorry for the unclear expression of our motivation in our previous version. It has now been articulated, as shown in Figure 1. We can summarize our motivation as reducing many-to-many correspondence between top convolutional filters and classes, so as to obtain class-related representation that facilitates human’s understanding and applications.
>
> In a vanilla CNN, each filter in the top convolutional layer corresponds to multiple classes, since it extracts a mixture of features from different classes, which is a symptom of filter ambiguity. Such an intricate many-to-many correspondence between filters and classes makes it hard for people to understand the intrinsic logic and further diagnose the networks. To reduce the filter ambiguity and force such a correspondence to be many-to-one, we propose LSG to regularize the training of CNNs which disentangles the filters and leads to class-specific representation. Our class-related representation is straightforward for human's understanding and we prove the superiority of this representation with exhaustive experiments on different application tasks including localization and adversarial samples detection, as in Section 4.4 and 4.5. of the revised version.
>
> Q: The proposed motivations partially misleading.
>
> A: As in the caption of Figure 1, our goal is to disentangle the filters in the last convolutional layer rather than all convolutional layers in general, because the last convolutional layer captures the most semantic and high-level information and deserves disentangling. We have revised the introduction to emphasize the top convolutional layer.
>
> Q: The advantage of enforcing the last layer filters to be class-specific is not clear.
>
> A: The advantage of class-specific filters in top convolutional layer is in two aspects. On the one hand, for CNN’s interpretability, a top convolutional filter focusing on one (or few) classes can extract more class-specific features or semantics. This facilitates understanding the inner logic of CNNs and diagnosing CNNs. On the other hand, for applications, class-specific filters  yield a better class-related feature representation. In a class-related feature representation, it’s easier to tell which class the image belongs to based on the channels with dominant activations. Our added experiments demonstrated that such representation significantly improves the performance of localization and adversarial samples detection.

---

> ### Author Response · Authors · 2019-11-15
> **Quantitative evaluation on localization and interpretability.**
>
>
> Q: Fig. 6 is only qualitative and there is no real localization measure.
>
> A: As you suggested, we conduct further quantitative evaluation with Avg IoU and AP now in Section 4.4. Referring to apply Network dissection (Bau, David, et al. 2017), Classification Activation Map (CAM) (Zhou, Bolei, et al. 2016) , we use both single filter’s feature map and CAMs to localize objects in each class. From Table 3 we can see that LSG significantly outperforms STD in terms of Avg-IoU and AP15. Moreover, the visualization of CAMs in Figure 7 also corroborates the better localization ability of our LSG.
>
> Q: However not much more interpretability is added.
>
> A: To further evaluate the enhanced interpretability, we supplement experiments on localization in Section 4.4. As a matter of fact, localization has played a crucial role in evaluating the interpretability of neural networks (Bau, David, et al. 2017) (Zhou, Bolei, et al. 2016) (Zhang, Quanshi, et al. 2018). Our increased performance on localization corroborates the better interpretability.
>
> ---
> Bau, David, et al. "Network dissection: Quantifying interpretability of deep visual representations." Proceedings of the IEEE Conference on Computer Vision and Pattern Recognition. 2017.
>
> Zhou, Bolei, et al. "Learning deep features for discriminative localization." Proceedings of the IEEE conference on computer vision and pattern recognition. 2016.
>
> Zhang, Quanshi, Ying Nian Wu, and Song-Chun Zhu. "Interpretable convolutional neural networks." Proceedings of the IEEE Conference on Computer Vision and Pattern Recognition. 2018.

---

> ### Author Response · Authors · 2019-11-15
> **Strengthening evidence in Table 1**
>
>
> Q: Evaluating the proposed approach on a single dataset and only one network is not enough for a clear evidence.
>
> A: Following your advice, we add ResNet152s trained on PASCAL VOC 2010 (preprocessed to be a classification dataset) in our revised version and report their metrics in Table 1. Besides, the ResNet152s are also used to quantitatively evaluate localization performance. The metrics of ResNet20 on CIFAR-10 and ResNet152s on PASCAL VOC both support the effectiveness of LSG training.
>
> Q: The obtained performance is below the standard performance of ResNet on CIFAR10 without a clear reason.
>
> A: The standard performance of ResNet20 on CIFAR-10 is 91.25% (He, Kaiming, et al. 2016), which indeed outperforms our results. However, our ultimate goal here is to provide a generic training strategy for better interpretability of CNNs. Therefore, we were not dedicated significantly to improving the classification accuracy of our model on CIFAR-10 using all kinds of optimization tricks, such as the fine-tuning of optimizer hyper-parameters, the initial learning rate, or the learning rate schedule. In addition, we use the SGD optimizer that is slower in making the model converge that other optimizers and only train for 150 epochs, so a relatively modest classification accuracy is not surprising. Nevertheless, we ensure the consistency of all hyperparameters of the STD CNN and LSG CNN for absolutely fair comparison, which excludes any possible bias in the analysis of the model interpretability.
>
> ---
> He, Kaiming, et al. "Deep residual learning for image recognition." Proceedings of the IEEE conference on computer vision and pattern recognition. 2016.

---

> ### Author Response · Authors · 2019-11-15
> **Replying misc comments**
>
>
> Q: How is redundancy reduced?
>
> A: In section 4.3 (revised version) we have added detailed analysis on filter redundancy.
> In section 4.3,  we began with filter orthogonality analysis, which  demonstrates that inter-class filters’s weights tend to be orthogonal. We have given an intuitive explanation for the orthogonality in the revised version. Then in further analysis we observed in Figure 7 (e) that LSG significantly reduces the redundant weak correlation between most filter pairs, which are mostly for different classes.
>
> It is a natural opinion that the reduction of filter redundancy is brought by the orthogonality of inter-class filters. Because for a set of filter groups orthogonal to each other, a filter in any group can not be a linear representation with the filters from other groups. This directly avoids redundant filter across groups. Besides, (Prakash, Aaditya, et al. 2019) conducted experiments to reduce filter redundancy by enforcing  filter orthogonality, which also verifies the opinion above.
>
> Q：Instead of evaluating the activation of the classification layer, it is possible to check the activation of the filters of the last convolutional layers.
>
> A: In our paper the focus is always on the top convolutional layer rather than the classification layer. We have made the concentration on the top convolutional layer more obvious in our new introduction. Besides, all the experiments have always been revolving around the top convolutional layer.
>
> Q: Another way to enforce a similar pattern on the gating function would be by maximizing the mutual information between selected filters and classes.
>
> A: Thanks for your advice. We like this idea! And we plan to discuss this in our future work, just like what infoGAN does.
>
> ---
>
> Prakash, Aaditya, et al. "RePr: Improved Training of Convolutional Filters." Proceedings of the IEEE Conference on Computer Vision and Pattern Recognition. 2019.

---

### Decision · Program_Chairs · 2019-12-19

**Decision:**

Reject

**Comment:**

The paper proposes a method to make the filters of the last conv layer more class-specific. The motivation for this is to improve upon the interpretability of the CNN, which is empirically shown by comparing the class activation maps (CAMs) of regular CNN and the proposed LSG-CNN. While the idea is interesting, one of the concerns from reviewers is about limited applicability of the method, at least the way it is shown in experiments -- a concern that I tend to agree with. As primary goal of the work is improving interpretability of CNNs, authors should test LSG-CNN with some more recent methods for producing the saliency maps other than CAM to convincingly establish the value of the method. Authors also mention lack of hyperparameter tuning and the use of SGD with limited training epochs as a reason for the drop in accuracy. It will be worth spending some effort so the accuracy matches the standard benchmarks -- this will help in arguing more convincingly about practical benefit of the method.